# REPRESENTATION LEARNING WITH MULTISETS

## ABSTRACT

We study the problem of learning permutation invariant representations that can capture "flexible" notions of containment. We formalize this problem via a measure theoretic definition of multisets, and obtain a theoretically-motivated learning model. We propose training this model on a novel task: predicting the size of the symmetric difference (or intersection) between pairs of multisets. We demonstrate that our model not only performs very well on predicting containment relations (and more effectively predict the sizes of symmetric differences than DeepSets-based approaches with unconstrained object representations), but that it also learns meaningful representations.

## 1 INTRODUCTION

Tasks for which the input is an unordered collection, i.e. a *set*, are ubiquitous and include multiple-instance learning Ilse et al. (2018), point-cloud classification Zaheer et al. (2017); Qi et al. (2017), estimating cosmological parameters Zaheer et al. (2017); Ravanbakhsh et al. (2016), collaborative filtering Hartford et al. (2018), and relation extraction Verga et al. (2017); Rossiello et al. (2019). Recent work has demonstrated the benefits of permutation invariant models that have inductive biases well aligned with the set-based input of the tasks (Ilse et al., 2018; Qi et al., 2017; Zaheer et al., 2017; Lee et al., 2019).

The containment relationship between sets — and intersection more generally — is often considered as a measure of relatedness. For instance, when comparing the keywords for two documents, we may wish to model that $\{\texttt{currency}, \texttt{equilibrium}\}$ describes a more specific set of topics than (i.e. is "contained" in) $\{\texttt{money}, \texttt{balance}, \texttt{economics}\}$. The containment order is a natural partial order on sets. However, we are often interested not in sets, but *multisets*, which may contain multiple copies of the same object; examples include bags-of-words, geo-location data over a time period, and data in any multiple-instance learning setting (Ilse et al., 2018). The containment order can be extended to multisets. Learning to represent multisets in a way that respects this partial order is a core representation learning challenge. Note that this may require modeling not just exact containment, but relations that consider the relatedness of individual objects. We may want to learn representations of the multisets' *elements* which induce the desired multiset relations. In the aforementioned example, we may want $\texttt{money} \approx \texttt{currency}$ and $\texttt{balance} \approx \texttt{equilibrium}$.

Previous work has considered modeling hierarchical relationships or orderings between pairs of individual items (Ganea et al., 2018; Lai and Hockenmaier, 2017; Nickel and Kiela, 2017; Suzuki et al., 2019; Vendrov et al., 2015; Vilnis et al., 2018; Vilnis and McCallum, 2015; Li et al., 2019; Athiwaratkun and Wilson, 2018). However, this work does not naturally extend from representing individual items to modeling relations between multisets via the elements' learned representations. Furthermore, we may want to consider richer information about the relationship between two multisets beyond containment, such as the size of their intersection.

In this paper, we present a measure-theoretic definition of multisets, which lets us formally define the "flexible containment" notion exemplified above. The theory lets us derive method for learning representations of multisets and their elements, given the relationships between pairs of multisets — in particular, we propose to use the sizes of their symmetric differences or of their intersections. We learn these representations with the goal of predicting the relationships between unseen pairs of multisets (whose elements may themselves have been unseen during training). We prove that this allows us to predict containment relations between unseen pairs of multisets. We show empirically that the theoretical basis of our model is important for being able to capture these relations, comparing our

approach to DeepSets-based approaches (Zaheer et al., 2017) with unconstrained item representations. Furthermore, we demonstrate that our model learns "meaningful" representations.

## 2 RELATED WORK

### 2.1 SET REPRESENTATION

Qi et al. (2017) and Zaheer et al. (2017) both explore learning functions on sets. Importantly, they arrive at similar theoretical statements about the approximation of such functions, which rely on permutation invariant pooling functions. In particular, Zaheer et al. (2017) show that any set function $f(A)$ can be approximated by a model of the form $\rho\left(\sum_{a \in A} \phi(a)\right)$ for some learned $\rho$ and $\phi$, which they call DeepSets. They note that the sum can be replaced by a max-pool (which is essentially the formulation of Qi et al. (2017)), and observe empirically that this leads to better performance.[1] More recently, there has been some very interesting work on leveraging the relationship between sets. Probst (2018) proposes a set autoencoder, while Skianis et al. (2019) learn set representations with a network that compares the input set to trainable "hidden sets." However, both these approaches require solving computationally expensive matching problems at each iteration.

### 2.2 ORDERS AND HIERARCHIES

Vendrov et al. (2015) and Ganea et al. (2018) seek to model partial orders on objects via geometric relationships between their embeddings — namely, using cones in Euclidean space and hyperbolic space, respectively. Nickel and Kiela (2017) use a similar idea to embed hierarchical network structures in hyperbolic space, simply using the hyperbolic distance between embeddings. These approaches are unified under the framework of "disk embeddings" by Suzuki et al. (2019). The idea is to map each object to the product space $X \times \mathbb{R}$, where $X$ is a (pseudo-)metric space. This mapping can be expressed as $A \mapsto (f(A), r(A))$, and it is trained with the objective that $A \preceq B$ if and only if $d_X(f(A), f(B)) \leq r(B) - r(A)$. An equivalent statement can be made for multisets (see Proposition 3.2.4).

Other work has taken a probabilistic approach to the problem of representing hierarchical relationships. Lai and Hockenmaier (2017) attempt to formulate the Order Embeddings of Vendrov et al. (2015) probabilistically, modeling joint probabilities as the volumes of cone intersections. Vilnis et al. (2018) represent entities as "box embeddings," or rectangular volumes, where containment of one box inside another models order relationships between the objects. (Marginal and conditional probabilities can be computed from intersections of boxes.) Vilnis and McCallum (2015) propose modeling words as Gaussian distributions in order to capture notions of entailment and generality, and this work has been extended to mixtures of Gaussians by Athiwaratkun and Wilson (2017).

### 2.3 FUZZY- AND MULTI- SETS

The theory of fuzzy sets can be traced back to Zadeh (1965). A fuzzy set $A$ of objects from a universe $\mathcal{U}$ is defined via its *membership function* $\mu_A : \mathcal{U} \to [0, 1]$. Fuzzy set operations — such as intersection — are then defined in terms of this function. In modern fuzzy set theory, intersection is usually defined via a *t-norm*, which is a function $T : [0, 1]^2 \to [0, 1]$ satisfying certain properties. The intersection of two fuzzy sets $A$ and $B$ is defined via the membership function $\mu_{A \cap B}(x) = T(\mu_A(x), \mu_B(x))$. (More in-depth background, including the defining properties of t-norms, is provided in Appendix B.) There is also more recent literature on extending fuzzy set theory to multisets (Casasnovas and Mayor, 2008; Miyamoto, 2000), using a membership function of the form $\mu_A : \mathcal{U} \times [0, 1] \to \mathbb{N}$, where $\mu_A(x, \alpha)$ is the number of appearances in $A$ of an object $x$ with membership $\alpha$.

---

[1] We believe there is an interesting theoretical distinction worth noting here, which may help explain this observation. Namely, max-pooling is *idempotent*, meaning that repeatedly pooling a representation with itself does not change the result. On the other hand, summation does not have this property, and so repeated copies of an element are reflected in the result. In this way, DeepSets (with the sum rather than max-pool) is in fact modeling *multisets* rather than sets, which depending on the application may be undesirable.

## 3 PROBLEM FORMULATION

Our goal is to learn to represent and predict a notion of containment between multisets. We begin with a brief motivating example, and then move on to provide the formalization of the problem. Recall our example, where for $A = \{\texttt{currency}, \texttt{equilibrium}\}$ and $B = \{\texttt{money}, \texttt{balance}, \texttt{economics}\}$, we have a sense in which $A$ is "contained" in $B$. After seeing many such example pairs of multisets, we want to be able to deduce that for $A' = \{\texttt{currency}, \texttt{food}\}$ and $B' = \{\texttt{money}, \texttt{food}\}$, the relation $A' \subseteq B'$ holds.

### 3.1 MULTISETS

In general, there exists a universe $\Omega$ of objects (in our above example, words). We let $\Omega^*$ denote the set of all multisets of objects from $\Omega$, formally defined as follows.

**Definition 3.1.1** *A multiset $A$ is defined by a its* membership function $m_A : \Omega \to M$, *where $M \subseteq \mathbb{R}_+$ is a subset of the non-negative reals, and $m_A$ maps each object to the "number of times" it occurs in $A$.*

The choice of $M$ dictates the kind of multiset $A$ is. In particular, $M = \{0, 1\}$ gives classical sets, and $M = \mathbb{N}$ gives the traditional notion of multiset — a set which may contain multiple copies of the same object. If $M = \mathbb{R}_+$, then we call $A$ a "fuzzy multiset."[2]

The cardinality (or size) of a multiset is defined with respect to a measure $\lambda$ on $\Omega$.

**Definition 3.1.2** *The* cardinality *of a multiset $A$ is* $|A| = \int_\Omega m_A(x) d\lambda(x)$.

We will always fix some measure $\lambda$ on $\Omega$ (which may be called the *dominating measure*) and take all cardinalities with respect to $\lambda$. Note that we can always view $m_A$ as a *density* (i.e. the Radon-Nikodym derivative) of some measure $\mu_A$ on $\Omega$ with respect to $\lambda$. We can thus identify a multiset $A$ with the measure $\mu_A$ on $\Omega$, and write $|A| = \mu_A(\Omega)$.

In the case that the universe $\Omega$ is countable, we simply let $\lambda$ be the counting measure, in which case the cardinality of any multiset $A$ is $|A| = \sum_{x \in \Omega} m_a(x)$.

All the usual operations on pairs of multisets are defined in terms of their membership functions.

**Definition 3.1.3** *For two multisets $A$ and $B$, their* intersection $A \cap B$, *union* $A \cup B$, *sum* $A + B$, *difference* $A - B$, *and* symmetric difference $A \triangle B$, *are multisets given by the following memberships functions, respectively:*

- $m_{A \cap B}(x) = \min\{m_A(x), m_B(x)\}$
- $m_{A \cup B}(x) = \max\{m_A(x), m_B(x)\}$
- $m_{A+B}(x) = m_A(x) + m_B(x)$

- $m_{A \setminus B}(x) = \max\{m_A(x) - m_B(x), 0\}$
- $m_{A \triangle B}(x) = |m_A(x) - m_B(x)|$

These definitions are standard for multisets with both whole-number and real-valued memberships (Casasnovas and Mayor, 2008; Miyamoto, 2000; Blizard, 1989). To those familiar with fuzzy set theory, it should immediately stand out the intersection and union are given by the standard T-norm and T-conorm (functions used to define these operations on fuzzy sets; see Appendix B). This means that our definition of fuzzy multiset contains a copy of fuzzy set theory. Unfortunately, there is no intuitive way to use other T-norms in order to define multiset operations. (For intuition on the above operations and why this is the case, see Appendix A.)

Finally, for multisets, containment is formally defined as follows.

---

[2]Note that this is not the same formulation of "fuzzy multisets" usually given in literature (Casasnovas and Mayor, 2008; Miyamoto, 2000). However, this formulation will be much more easily amenable to the machine-learning setting. Our notion here is also more closely related to the "real-valued multisets" of Blizard (1989), although the author approaches the subject from a standpoint of formal logic and axiomatic set theory.

**Definition 3.1.4** *For two multisets $A \in \Omega^*$ and $B \in \Omega^*$, we say that $B$ contains $A$, or $A \subseteq B$, if and only if $m_A(x) \leq m_B(x)$ for all $x \in \Omega$.*

Note however that as demonstrated by our motivating example above, for two multisets $A$ and $B$, we want to have a more "flexible" notion than $A \subseteq B$. (It is not actually the case that $\{\texttt{currency}, \texttt{equilibrium}\} \subseteq \{\texttt{money}, \texttt{balance}, \texttt{economics}\}$.) We will now formally provide a structure allowing for this flexibility.

## 3.2 A "FLEXIBLE" NOTION OF CONTAINMENT

Let us first make two observations. Firstly, the desired flexibility will depend on some notion of "similairty" between the objects in $\Omega$. Secondly, this similarity must be externally provided by our observations of these "subset" relations. In our example above, we had a sense that $\texttt{money} \approx \texttt{currency}$ and $\texttt{balance} \approx \texttt{equilibrium}$, *because* we observed that $A$ is "contained" in $B$ (perhaps along with many other similar examples). We now formalize this idea.

**Definition 3.2.1** *For two universes $\Omega$ and $\mathcal{U}$, a map $T : \Omega^* \to \mathcal{U}^*$ is a called a* multiset transformation *from $\Omega$ to $\mathcal{U}$.*

The idea is that there exists some multiset transformation $T$ from $\Omega$ to $\mathcal{U}$, but we may not observe the structure of $T$ or this new universe $\mathcal{U}$. However, we indirectly observe this structure, because our notion of subsets will be taken in $\mathcal{U}^*$ rather than in $\Omega^*$. In particular, in our example above, the sense in which $A$ is "contained" in $B$ is that $T(A) \subseteq T(B)$.

The simplest example of such a setting, on which we focus, is when $\mathcal{U} = \{1, \dots, k\} = [k]$. That is, $T$ maps each multiset in $\Omega^*$ to a multiset of numbers from 1 to $k$. These numbers can be thought of as "tags," "classes," or "labels," meaning that each multiset has associated to it some tags, each of which may occur more than once. Say for our running example, $T(A) = \{1, 2\}$ and $T(B) = \{1, 2, 3\}$. We then obtain that $T(A) \subseteq T(B)$, as desired.

The example mapping $T$ above is suggestive. It suggests a category of such $T$ functions that are commonly useful: when each object in $\Omega$ is itself associated with a "tag" in $\mathcal{U}$. Here, $\texttt{money}$ and $\texttt{currency}$ both are associated to tag 1, $\texttt{balance}$ and $\texttt{equilibrium}$ are both associated to tag 2, and $\texttt{economics}$ to tag 3. Our map $T$ is induced by this element-wise mapping. We formalize this notion as follows.

**Definition 3.2.2** *We call a function $t : \Omega \to \mathcal{U}$ a* universe transformation. *Each universe transformation $t$ induces a* pushforward multiset transformation $T$, *where the membership function of $T(A)$ is $m_{T(A)}(y) = \int_{t^{-1}(y)} m_A(x) d\lambda(x)$.*

**Proposition 3.2.3** *Every pushforward multiset transformation $T$ preserves cardinalities; that is, for any $A$, we have $|A| = |T(A)|$.*

Note that if we view $A$ as the measure $\mu_A$ on $\Omega$, then the measure $\mu_{T(A)}$ on $\mathcal{U}$ is the honest-to-goodness pushforward measure $\mu_A \circ t^{-1}$. The above result follows easily.

Let's summarize where we are so far. We observe some relations between pairs of multisets over $\Omega$. We also assume there is a multiset transformation $T$ from $\Omega$ to some "latent" universe $\mathcal{U}$, and that our observed relations are explained by relations that hold in $\mathcal{U}^*$. In general, we wish to understand the structure of $T$, in order to predict similar relations for unobserved pairs of multisets. We may also hope that in the process we learn something about the structure of $\Omega$, and that in general this learning process is feasible due to $\mathcal{U}$ being much smaller or simpler than $\Omega$. For example, we might assume that $T$ is in fact a pushforward transformation induced by an unobserved labeling of the elements in $\Omega$ by elements in $\mathcal{U} = [k]$.[3]

However, there is a problem with this setup: it is unlikely that for two multisets $A \in \Omega^*$ and $B \in \Omega^*$, we have either $T(A) \subseteq T(B)$ or $T(B) \subseteq T(A)$. However, there are richer relations that can exist

---

[3]It is worth noting that there do exist examples where each object in $\Omega$ does not obviously have an associated tag in $[k]$, such as the RCV1 dataset of Lewis et al. (2004), which consists of documents — each of which can be thought of as a multiset of words — together with collections of categories for each document.

between two multisets than just containment, and which can be observed for any such pair. For example, regardless of whether either of $T(A)$ or $T(B)$ is a subset of the other, we can ask about how much they overlap — i.e. the size of either their intersection $|T(A) \cap T(B)|$, or of their symmetric difference, $|T(A) \triangle T(B)|$. Note that if we also know their sizes $|T(A)|$ and $|T(B)|$, then we can know whether one contains the other (Theorem 3.2.6). This follows directly from the following: (1) that the size of the symmetric difference gives rise to a (pseudo-)metric on multisets, which can then be used to express a "disk embedding" inequality (Suzuki et al., 2019) relating containment to cardinalities (Proposition 3.2.4); and (2), the sizes of the symmetric difference and intersection are related via the sizes of the multisets themselves (Lemma 3.2.5). See Appendices C and D for proofs; note also that for each of the following three statements, we assume $A$ and $B$ are multisets over the same universe.

**Proposition 3.2.4** *For any two multisets $A$ and $B$, $A \subseteq B$ if and only if $|A \triangle B| \leq |B| - |A|$.*

**Lemma 3.2.5** *For any two multisets $A$ and $B$, $|A \triangle B| = |A| + |B| - 2|A \cap B|$.*

**Theorem 3.2.6** *Given $A$ and $B$ (whose cardinalities we can calculate), it suffices to know either $|A \cap B|$ or $|A \triangle B|$ to conclude whether $A \subseteq B$ and whether $B \subseteq A$.*

Theorem 3.2.6 thus motivates the use of either $|T(A) \cap T(B)|$ or $|T(A) \triangle T(B)|$ to learn about containment, in the case where $T$ preserves cardinalities. In particular, we then know the cardinalities $|T(A)| = |A|$ and $|T(B)| = |B|$, and thus either $|T(A) \cap T(B)|$ or $|T(A) \triangle T(B)|$ is sufficient to deduce the containment relations between $T(A)$ and $T(B)$. We therefore see that we can use a training signal more readily available than binary yes-no containment — measurements of overlap between multisets — to still learn to predict containment relations. Thus, the problem we will be solving here is learning to predict either $|T(A) \triangle T(B)|$ or $|T(A) \cap T(B)|$ from examples. Importantly, the error on these predictions will indicate how well we learned to capture our "flexible" notion of containment.

### 3.3 THE LEARNING TASK

Formally, our learning task will therefore be as follows.

There exists a universe $\Omega$, which we assume for practical purposes can be embedded in $\mathbb{R}^d$ for some known $d$. There is also some latent universe $\mathcal{U} = [k]$ together with an unknown multiset transformation $T : \Omega^* \to \mathcal{U}^*$. We will assume that $T$ preserves cardinalities — in practice this means either that $T$ is a pushfoward transformation induced by some $t : \Omega \to \mathcal{U}$, or an "expectation transformation," which we define in Section 4.2. We then observe samples $(A, B)$ from a training distribution $D$ over pairs of multisets in $\Omega^*$. For practical reasons, our sampled multisets will whole-number multiplicities. For each such pair, we also observe the overlap via either $|T(A) \triangle T(B)|$ or $|T(A) \cap T(B)|$. Which of these is used is fixed beforehand for the entire task, and we assume this choice is known. (We test both choices in our experiments.) Our assumption that $T$ preserves cardinality is important, because together with these observations, it allows us to conclude whether $T(A)$ is a subset of $T(B)$. We then pick a hypothesis target universe $\hat{\mathcal{U}} = [\hat{k}]$, and we let $\hat{k} = k$ if $k$ is known. (Experimentally, we examine the cases $\hat{k} < k$ and $\hat{k} = k$.) Finally, our goal is to learn a model — i.e. a map $\hat{T} : \Omega^* \to \hat{\mathcal{U}}^*$ — that minimizes squared error in the predicted overlaps. That is, we learn $\hat{T}$ in order to minimize the appropriate choice of the following two losses:

$$\mathcal{L}_\triangle = \mathbb{E}_{A,B \sim D} \left[ \left( \left| \hat{T}(A) \triangle \hat{T}(B) \right| - |T(A) \triangle T(B)| \right)^2 \right]$$

$$\mathcal{L}_\cap = \mathbb{E}_{A,B \sim D} \left[ \left( \left| \hat{T}(A) \cap \hat{T}(B) \right| - |T(A) \cap T(B)| \right)^2 \right].$$

### 4 MODEL DEFINITION

Having formulated our learning task, we now define our learnable model $\hat{T} : \Omega^* \to \hat{\mathcal{U}}^*$. In order to do so, we want our model to give us "representations" of the multisets in $\hat{\mathcal{U}}^*$ in the most common machine-learning sense — i.e. vectors in some Euclidean space. We begin this section by defining how we obtain and use such representations, and then conclude by defining our model $\hat{T}$ itself that gives us these representations.

### 4.1 REPRESENTATION OF MULTISETS

**Definition 4.1.1** *For any universe $\hat{\mathcal{U}}$, a $d$-dimensional representation function is a map $\Psi : \hat{\mathcal{U}}^* \to \mathbb{R}^d$. For a multiset $S \in \hat{\mathcal{U}}^*$, we call $\Psi(S)$ the* representation of $S$.

In general, we want our representations of multisets to be "useful," in the sense that we can use them to perform common operations — such as those in Definition 3.1.3. More importantly for our task, we need to be able to calculate the size of either the symmetric difference or the intersection of two multisets. Our choice of target universe $\hat{\mathcal{U}} = [\hat{k}]$ gives us such a representation function.

**Definition 4.1.2** *Let $\hat{\mathcal{U}}$ be the finite universe $[\hat{k}]$. The* natural representation function $\Psi_{\hat{k}} : \hat{\mathcal{U}}^* \to \mathbb{R}^{\hat{k}}$ *is the map $S \mapsto [m_S(1), \ldots, m_S(\hat{k})]$.*

This should be an intuitive concept. For example, the natural representation function for classical sets gives the familiar indicator vector representation $\Psi_{\hat{k}}(S) = [\mathbf{1}_{1 \in S}, \ldots, \mathbf{1}_{\hat{k} \in S}]$. Furthermore, we get "usefullness" of these representations for free, since all operations defined via membership functions (e.g. those in Definition 3.1.3) can be performed coordinate-wise. Furthermore, the cardinality of a multiset $S \in \hat{\mathcal{U}}^*$ is given by thus sum of the entries in $\Psi_{\hat{k}}(S)$. Together with the non-negativity of membership functions, this gives us the following. (As these two results are essentially immediate, we omit their proofs.)

**Lemma 4.1.3** *For any multiset $S \in [\hat{k}]^*$, $|S| = ||\Psi_{\hat{k}}(S)||_1$.*

**Proposition 4.1.4** *For any two multisets $R$ and $S$ over $[\hat{k}]$, we have $|R \triangle S| = ||\Psi_{\hat{k}}(R) - \Psi_{\hat{k}}(S)||_1$ and $|R \cap S| = \min\{\Psi_{\hat{k}}(R), \Psi_{\hat{k}}(S)\}$, where the minimum is applied coordinate-wise.*

We thus use the natural representation function on $\hat{\mathcal{U}}$ to train our model. We note that Proposition 4.1.4 could provide a reason to prefer the size of the symmetric difference over the size of the intersection as the training signal. The reasoning is that $\ell_1$-distance has a gradient which depends on both the representations $\Psi_{\hat{k}}(R)$ and $\Psi_{\hat{k}}(S)$ in each coordinate (except at 0), while the coordinate-wise minimum can only depend on one of the representations in each coordinate. We test this idea in our experiments.

### 4.2 THE LEARNABLE MODEL

Recall that the unobserved multiset transformation $T : \Omega^* \to \mathcal{U}^*$ preserves cardinalities. In particular, suppose for the purpose of exposition that $T$ is the pushforward transformation induced by some labeling $t : \Omega \to \mathcal{U}$. We both want our hypothesis class of models $\hat{T}$ to contain all such pushforward multiset transformations, and to potentially be restricted to those $\hat{T}$ which preserve cardinalities. Unfortunately, we cannot directly learn over the set of all pushforward transformations, as this is equivalent to learning the correct discrete labeling $t : \Omega \to [k]$, which is both a hard and non-differentiable problem.[4] Instead, we take a probabilistic approach.

**Definition 4.2.1** *For two universes $\Omega$ and $\hat{\mathcal{U}}$, a* probabilistic universe transformation *is a map $\ell : \Omega \to \Delta(\hat{\mathcal{U}})$, where $\Delta(\hat{\mathcal{U}})$ is the space of probability measures on $\hat{\mathcal{U}}$.*

As we will see, probabilistic universe transformations to $[\hat{k}]$ have the advantage of being smoothly parametrizable. In analogy to the pushforward multiset transformation induced by a $t : \Omega \to \mathcal{U}$, we leverage our probabilistic transformation above to define a different kind of induced multiset transformation.

**Definition 4.2.2** *Let $\ell : \Omega \to \Delta(\hat{\mathcal{U}})$ be a probabilistic universe transformation. The* expectation multiset transformation $L : \Omega^* \to \hat{\mathcal{U}}^*$ *is defined to be the map $A \mapsto \mathbb{E}_{P \sim (\mu_A \circ \ell^{-1})}[P]$.*

---

[4]In fact, we only care about learning such a labeling up to permutation (as permutations of labels cannot affect containment relations), but this is still a hard problem.

We first note that $L$ is well defined, in the sense that $L(A)$ is always a valid measure on $\hat{\mathcal{U}}$. This can easily be seen by re-writing the expression as follows:

$$\mathbb{E}_{P \sim (\mu_A \circ \ell^{-1})}[P] = \int_{\Delta(\hat{\mathcal{U}})} P \, d \left( \mu_A \circ \ell^{-1} \right)(P) = \int_{\Omega} \ell(x) \, d\mu_A(x).$$

$L(A)$ has a natural interpretation, as the expected multiset obtained by sampling an element of $\hat{\mathcal{U}}$ for each $x \in A$ according to its corresponding distribution $\ell(x)$ (with contribution weighted by $m_A(x)$). Additionally, we have the desirable property that $L$ preserves cardinalities (see Appendix E for proof).

**Theorem 4.2.3** *Any expectation multiset transformation $L : \Omega^* \to \hat{\mathcal{U}}^*$ preserves cardinalities, i.e. for any $A \in \Omega^*$, $|A| = |L(A)|$.*

We will thus let our learned model $\hat{T} : \Omega^* \to \hat{\mathcal{U}}^*$ be an expectation transformation $L$ induced by some probabilistic universe transformation $\ell$. Note that we expect to be able to learn not only in the case where the unknown $T$ is a pushforward transformation, but in fact when $T$ itself is some expectation transformation (although we do not test the latter experimentally.)

## 4.3 PUTTING IT ALL TOGETHER

The outstanding question is: how do we parametrize our representations $\Psi_{\hat{k}}(L(A))$ for any given $A \in \Omega^*$? By the definition of the natural representation function $\Psi_{\hat{k}}$, we can write the $i$-th component of the representation of $L(A)$ as

$$m_{L(A)}(i) = \int_{\Omega} (\ell(x))(\{i\}) \, d\mu_A(x) = \int_{\Omega} m_{\ell(x)}(i) \, d\mu_A(x) = \int_{\Omega} \Psi_{\hat{k}}(\ell(x))_i \, d\mu_A(x),$$

where the last two equalities come from viewing $\ell(x)$ as a general measure and thus multiset on $\hat{\mathcal{U}}$. Now, assuming that $A$ is in fact one of the multisets sampled from our training distribution $D$, we know that $A$ has whole-number multiplicities. (This will also be the case during evaluation, and thus in fact for any multiset we are trying to represent.) The above can then by simply written as $\Psi_{\hat{k}}(L(A))_i = \sum_{x \in A} \Psi_{\hat{k}}(\ell(x))_i$, where each $x$ occurs in the sum $m_A(x)$ times. More simply, we have $\Psi_{\hat{k}}(L(A)) = \sum_{x \in A} \Psi_{\hat{k}}(\ell(x))$.

Since each $\ell(x)$ is just a distribution over $\hat{k}$ elements, it suffices to learn a map from $\Omega$ to the probability simplex in $\mathbb{R}^{\hat{k}}$ — the non-negative vectors whose components sum to 1. Recalling that we assumed our input universe $\Omega$ consists of vectors in $\mathbb{R}^d$, we pick a favorite object-featurization network $\phi : \mathbb{R}^d \to \mathbb{R}^{\hat{k}}$. We then guarantee than we obtain a point in the probability simplex by taking $\phi(a)$ to $\frac{f(\phi(a))}{||f(\phi(a))||_1}$, where $f : \mathbb{R} \to \mathbb{R}_+$ is a function applied component-wise. For differentiability, we choose the softplus function $f(x) = \log(1 + e^x)$. Our complete model is thus the representation $\Psi(A) = \sum_{x \in A} \frac{f(\phi(a))}{||f(\phi(a))||_1}$. Our losses, in terms of these representations, are

$$\mathcal{L}_{\triangle} = \mathbb{E}_{A,B \sim D} \left[ (||\Psi(A) - \Psi(B)||_1 - |T(A) \triangle T(B)|)^2 \right]$$

$$\mathcal{L}_{\cap} = \mathbb{E}_{A,B \sim D} \left[ (\min\{\Psi(A), \Psi(B)\} - |T(A) \cap T(B)|)^2 \right].$$

## 5 EXPERIMENTS

We begin here with an overview what we want to test about our model. In Section 5.1 we move on to describe our training and evaluation procedures. The experimental results themselves follow.

A clear question to seek the answer to empirically is whether the size of the symmetric difference or of the intersection works better in practice. (Recall that the symmetric difference may have more informative gradients, possibly leading to better learning and performance.) We thus compare these two approaches, both in terms of the error on the respective tasks themselves, and in terms of the error on predicting containment. More generally, the theory motivating our model suggest that there

is a delicate balance in the properties that make the model well-posed.[5] We tackle this idea from two directions.

First, we ask how important is the precise definition of our model, $\Psi(A) = \sum_{x \in A} \frac{f(\phi(a))}{||f(\phi(a))||_1}$. An obvious baseline to compare against is $\Psi(A) = \sum_{x \in A} \phi(x)$, which should help us answer the question of how important it is that each object is mapped to a point in the probability simplex. Looking at this formulation, an immediate connection one might make is to the DeepSets model of Zaheer et al. (2017): $\Psi(A) = \rho_1(\sum_{x \in A} \phi(x))$ for some learnable function $\rho_1$. The authors prove this model can learn any permutation invariant function — e.g. the size of the intersection or symmetric difference of two multisets. We thus use both the models above as baselines in all our experiments, calling the former "unrestricted multisets" and the latter "DeepSets"

The second category of question we ask here is whether we gain anything from our construction of the multiset operations on representations. We tackle this question replacing the terms $||\Psi(A) - \Psi(B)||_1$ and $\min\{\Psi(A)l\Psi(B)\}$ in our losses with $\rho_2(\Psi(A) + \Psi(B))$ for a learnable function $\rho_2$. The intuition here is that this new prediction is in fact a second DeepSets model trying to learn of our prediction functions — where we choose DeepSets because both intersection and symmetric difference are permutation invariant (i.e. commutative). This setting will be called the "learned operation" setting. We further test whether our parametrizations of the multisets operations are somehow intrisically good via a scheme of "cross-wiring" them — using one for a task where we should use the other — on which we elaborate in Section 5.4.

Finally, we will also examine the learned representations $\phi(x)$ of elements $x \in \Omega$.

## 5.1 TRAINING AND EVALUATION PROCEDURES

We use MNIST (LeCun, 1998) as our dataset. The training set consists of 60,000 handwritten images of digits, and the test set of 10,000.

We train all the models on $3 \times 10^5$ training pairs of multisets $(A, B) \in \Omega$. At each iteration of training, both $A$ and $B$ are generated randomly, as follows. First a whole-number cardinality is uniformly sampled in some chosen range — in our experiments we use $[2, 5]$. (We exclude singleton sets to ensure that the models aren't just learning from comparing pairs of singletons.) Once the cardinality is chosen, then that number of images $x \in \Omega$ is then chosen uniformly at random (with replacement) from the training set. The multiset representation is calculated as usual, via one of the representation functions $\Psi$ defined above, and the predicted cardinality of the symmetric difference or intersection is then calculated using these representations. The value to be predicted is calculated directly from the labels of the images in the multisets — e.g. if $A$ is two images of ones, and $B$ is a one and a three, the target value will be 1 for intersection, and 2 for symmetric difference. The squared error is minimized using Adam (Kingma and Ba, 2015), with the default parameters $\beta_1 = 0.9$ and $\beta_2 = 0.999$, and a learning rate of $5 \times 10^{-5}$. The learning rate was chosen by logarithmic grid search from 1 down to $5 \times 10^{-6}$, training on up to $10^4$ pairs during the search. (All models performed best with the chosen learning rate — or at least no worse than any of the other learning rates.) Given this learning rate, we chose to train the models for $3 \times 10^5$ iterations, finding that almost all of the models converged by this point.

Evaluation is performed similarly to training, with the addition of multiset sizes uniform on $[2, 20]$, and with images sampled from the test set. Importantly, this means that the none of the images seen during training appear during evaluation. Each model is evaluated on $3 \times 10^4$ such multiset pairs, and unless otherwise stated we let $\phi : \Omega \to \mathbb{R}^k$ (that is, $\hat{h} = k = 10$).

For the object featurizing function $\phi$, we use a variant of the LeNet-5 neural network (LeCun, 1998). Specifically, we adopt the same architecture as used by Ilse et al. (Ilse et al., 2018). (See Appendix F for network architectures, including those used for $\rho_1$ and $\rho_2$.)

---

[5]In particular, the restriction of the object representation to the probability simplex is important for our expectation-transformation-based model to be a well-defined and cardinality-preserving map between multisets; furthermore, the guarantee that the learned representations $\Psi(A)$ and $\Psi(B)$ for any pair $(A, B)$ preserve cardinalities is theoretically key both for our being able to use $|A|$ and $|B|$ (in addition to our observed symmetric differences or intersections) in order to deduce containment relations, as is the way in which we parametrize operations on multisets via operations on their representation. There are many possible directions of inquiry; we break apart the empirical question into two rough categories.

## 5.2 CARDINALITY PREDICTION

We examine the performance of six kinds of models — multisets, unrestricted multisets, and DeepSets, each with or without learned multiset operations — on the tasks of predicting cardinality, either of the symmetric difference of the intersection. We will refer here to Tables 1 (symmetric difference) and 2 (intersection), which report the mean absolute errors of the predictions. Within each of the tables, two patterns are immediately clear. First, a amount portion of the prediction error may be explained by whether the multiset operations are learned, or taken to be the theoretically-motivated parametrizations; the models with learned operations exhibit more than twice the prediction error. While this shows there is a benefit to using our theoretically-motivated definitions, it does not necessarily mean that our definitions are intrinsically or uniquely well-suited for the task. We will revisit this point later.

The second salient pattern is that as we move away from the expectation-transformation model (which we simply call "multisets" in our tables), first to the unrestricted multiset model, and then to DeepSets, there is a rapid decrease in performance (in some cases almost ten-fold). This suggests that the theory behind our model is indeed useful.

Finally, when we compare across the two tables — that is, compare cardinality prediction for symmetric difference and for intersection — we observe a surprisingly large gap in error. The error on intersection size prediction is consistently about twice as small as on the other task. It is worth noting that if anything, we expected an opposite effect. This gap is intriguing, and we believe that it should be explored further.

## 5.3 CONTAINMENT PREDICTION

We now compare the same models above on what is perhaps the more important task: prediction whether there exists a containment relation between $T(A)$ and $T(B)$ for some $A$ and $B$. In particular, for any such pair, we predict whether $T(A) = T(B)$, $T(A) \subsetneq T(B)$, $T(B) \subsetneq T(A)$, or there is containment relation (i.e. we treat this as a classification problem). We perform this prediction by relying of Theorem 3.2.6 (and the assumptions on our representations). In particular, for any pair $A$ and $B$, we predict the containment relation implied by Proposition 3.2.4, where we take the cardinality of the symmetric difference predicted by our model. (If the model predicts intersection, we just use Lemma 3.2.5 to go from one to the other.) Note that in this experiment we sample pairs $A$ and $B$ such that the probability of each kind of containment is essentially uniform. Referring to Tables 3 and 4, we observe almost exactly the same patters as above — with over 96% accuracy achieved by both multiset models. The one difference is that for the unnormalized model and the DeepSets model (with non-learned operations), the version of the models trained on the intersection task perform noticeably worse than the corresponding models trained on symmetric difference. Furthermore, on all other models, the performances are comparable. This further complicates the picture from above, as it suggests that while intersection may somehow be easier to learn to predict the cardinality of, perhaps the task itself is a worse way to capture containment relations.

## 5.4 "CROSS-WIRING" OPERATIONS

Motivated by our observation that models less closely aligned with our theory seems to perform worse, we devise a small test to see whether our symmetric difference and intersection cardinality operations are somehow intrinsic. To do so, we perform two experiments. First, we train our regular expectation-transformation-based multiset model to predict symmetric difference cardinality, but where it's prediction function is given by $\min\{\Psi(A), \Psi(B)\}$. Similarly, in the second experiment, we train the model to predict intersection cardinality, but where the prediction function is $||\Psi(A) - \Psi(B)||_1$. We observe that the former model achieves and mean-absolute error of 2.3710 on the test set, while the second achieves ones of 0.9929. There values are significantly higher than the errors achieved with the "correct" prediction functions, suggesting that there indeed a sense in which these are the "right" functions.

## 5.5 EXAMINING LEARNED REPRESENTATIONS

We finally turn to examining the object-representations learned by our model. As one would expect, the learned representations of objects are approximate the standard basis vectors (as shown in Figure

1 for $n = 3$). This suggests our expectation-transformation model is learning appropriate point-mass probabilities corresponding to each object's label in $\mathcal{U}$.

We also examine the case $\hat{k} < k$, which may occur when we don't know the true size of $\mathcal{U}$. Here, the "pinched" nature of the restricted representations may be undesirable (Figure 2a). This problem, of course, gets worse with the discrepancy between number of objects and dimension (Figure 2b). On the other hand, the unrestricted multiset model is able to learn more balanced-looking clusters. However, the clusters for $d = n$ appear slightly less well-separated (Figures 3 and 4). The DeepSets model didn't learn interpretable representations (Figure 4c). Furthermore, when we measure the accuracy of the regular multiset model on the containment prediction task above of each of the models, we obtain good results even when $\hat{k}$ is "too small." In particular, fixing $k = 5$: $\hat{k} = 5$ we obtain an accuracy of 0.9586, and $\hat{k} = 3$ gives 0.9045. This suggest that the representations learned are in fact robust to small discrepancies in dimension.

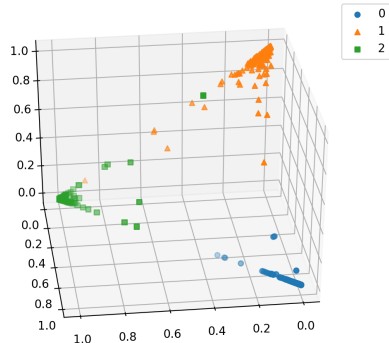

Figure 1: Three-dimensional representations of test-set MNIST images generated by the restricted multiset model trained on multisets of sizes $\in [2, 5]$; the model is trained on images of zeros, ones, twos.

## 6 CONCLUSION

We propose a novel task: predicting the size of either the symmetric difference of the intersection between pairs of multisets. We motivate this construction via a measure-theoretic notion of "flexible containment." We demonstrate the utility of this idea, developing a theoretically-motivated model that given only the sizes of symmetric differences between pairs of multisets, learns representations of such multisets and their elements. These representations allow us to predict containment relations with extremely high accuracy. Our model learns to map each type of object to a standard basis vector, thus essentially performing semi-supervised clustering. One interesting area for future theoretical work is understanding a related problem: clustering $n$ objects given multiset difference sizes. As a first step, we show in Appendix H that $n - 1$ specific multiset comparisons are sufficient to recover the clusters. We would also be curious to see if one can learn the latent multiset space $\mathcal{U}$.

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

## A    MULTISET OPERATIONS

Here we provide through examples an intuitive understanding of the binary multiset operations from Definition 3.1.3.

Consider the two (non-fuzzy) multisets, $A = \{1, 1, 1, 2, 2\}$ and $B = \{1, 1, 2, 3\}$. Their intersection should contain all their elements in common: $A \cap B = \{1, 1, 2\}$. That is, we take the minimum number of times each element appears in either $A$ or $B$, and that is the number of times the element appears in $A \cap B$. This straightforwardly gives us $m_{A \cap B}(x) = \min\{m_A(x), m_B(x)\}$.

Following similar reasoning, we can convince ourselves that multiset union should be defined as $m_{A \cup B}(x) = \max\{m_A(x), m_B(x)\}$. It is important to differentiate this from "multiset addition," which simply combines two multisets directly: $A + B = \{1, 1, 1, 1, 1, 2, 2, 2, 3\}$ for our example above, and in general $m_{A+B} = m_A(x) + m_B(x)$.

Multiset difference is a little harder to define. The main problem is that we cannot rely on a notion of "complement" for multisets. Instead, let us again try to reason by example. For our example multisets above, we have $A \setminus B = \{1, 2\}$. To arrive at this result, we remove from $A$ each copy of an element which also appears in $B$. Note that if $B$ had more of a certain element than $A$, that element would not appear in the final result. In other words, we are performing a subtraction of counts which is "glued" to a minimum value of zero. That is, $m_{A \setminus B}(x) = \max\{m_A(x) - m_B(x), 0\}$. We can further convince ourselves of the correctness of this expression by noting that we recover the identity $A \setminus (A \setminus B) = A \cap B$.

Finally, symmetric multiset difference can be defined using our expression for multiset difference, combined with either multiset addition or union. In particular, note that $A \triangle B = (A \setminus B) + (B \setminus A) =$

$(A \setminus B) \cup (B \setminus A)$ — addition and union both work because $(A \setminus B)$ and $(B \setminus A)$ are necessarily disjoint. This gives us:

$$m_{A \triangle B}(x) = \max\{m_A(x) - m_B(x), 0\} + \max\{m_B(x) - m_A(x), 0\} = |m_A(x) - m_B(x)|.$$

(The equation still holds if we replace the addition with a maximum.)

## B  FUZZY SETS

A fuzzy set $A$ over a universe $\Omega$ is given by a function $m_A : \Omega \to [0, 1]$. Intuitively, $m_A$ maps each $x \in \Omega$ to "how much of a member" $x$ is of $A$, on a scale from 0 to 1. With this simple idea, fuzzy set operations can be defined. This is traditionally done by leveraging element-wise fuzzy logical operations, which we define below.

**Definition B.0.1** *A* t-norm *is a function* $T : [0, 1]^2 \to [0, 1]$*, satisfying the following properties:*

- *Commutativity:* $T(a, b) = T(b, a)$

- *Monotonicity: If* $a \le c$ *and* $b \le d$*, then* $T(a, b) \le T(c, d)$

- *Associativity:* $T(a, T(b, c)) = T(T(a, b), c)$

- *1 is the identity:* $T(a, 1) = a$

T-norms generalize the notion of conjunction. Note that the above conditions imply that for any $a$, $T(a, 0) = 0$, and that $T(1, 1) = 1$. These two observations show that t-norms are "compatible" with classical, non-fuzzy logic — where we identify 0 with "false" and 1 with "true." The standard t-norm is $T(a, b) = \min\{a, b\}$.

**Definition B.0.2** *A* strong negator *is a strictly monotonic, decreasing function* $n : [0, 1] \to [0, 1]$ *such that* $n(0) = 1$*,* $n(1) = 0$ *and* $n(n(x)) = x$*.*

Unsurprisingly, strong negators generalize logical negation. The standard strong negator is $n(x) = 1 - x$.

**Definition B.0.3** *An* S-norm *(also called a* t-conorm*) is a function with the same properties as a t-norm, except that the identity element is 0.*

S-norms generalize disjunction. For every t-norm (and a given negator), we can define a *complementary s-norm*: $S(a, b) = n(T(n(a), n(b)))$. This is a generalization of De Morgan's laws. The standard s-norm, complementary to the $\min$ t-norm, is $S(a, b) = \max\{a, b\}$.

The membership function for the intersection of two fuzzy sets $A$ and $B$ is naturally defined as $\mu_{A \cap B}(x) = T(\mu_A(x), \mu_B(x))$ for a t-norm $T$. Similarly, the complement of a fuzzy set is given by $\mu_{\overline{A}}(x) = n(\mu_A(x))$ for a strong negator $n$, and the union of two fuzzy sets is given by $\mu_{A \cup B}(x) = S(\mu_A(x), \mu_B(x))$ for an s-norm $S$. Usually, we want $T$ and $S$ to be complementary with respect to $n$. Then, we can generalize all the usual set operations to fuzzy sets by combining the three basic operations above.

## C  PROOF OF PROPOSITION 3.2.4

We show that for any two multisets $A$ and $B$ over the same universe $\Omega$, $A \subseteq B$ if and only if $|A \triangle B| \le |B| - |A|$. In fact, noting that it is always the case that $|A \triangle B| \ge |B| - |A|$ (which we will not prove but is easy to show), the following proof shows this holds with equality.

*Proof.*  Let $\lambda$ be the dominating measure with respect to which the cardinalities are taken.

We first show the forward direction. Suppose $A \subseteq B$, that is, for every $x \in \Omega$, we have $m_A(x) \le m_B(x)$. The result follows directly (with equality):

$$\int_\Omega m_{A \triangle B}(x) \, d\lambda(x) = \int_\Omega |m_B(x) - m_A(x)| \, d\lambda(x) = \int_\Omega m_B(x) \, d\lambda(x) - \int_\Omega m_A(x) \, d\lambda(x).$$

For the converse direction, suppose on the other hand that $|A \triangle B| \leq |B| - |A|$. Now suppose for the sake of contradiction that for some $x^* \in \Omega$, we have $m_A(x^*) > m_B(x^*)$. Then $m_B(x^*) - m_A(x^*) < |m_B(x^*) - m_A(x^*)|$. But this implies that

$$|B| - |A| = \int_\Omega m_B(x) - m_A(x) \, d\lambda(x) < \int_\Omega m_B(x) - m_A(x) \, d\lambda(x) = |A \triangle B|,$$

which is a contradiction. ∎

## D  PROOF OF LEMMA 3.2.5

We show that for any two multisets $A$ and $B$ over the same universe $\Omega$, $|A \triangle B| = |A| + |B| - 2|A \cap B|$.

*Proof.*  Let $\lambda$ be the dominating measure with respect to which the cardinalities are taken. Let $\Omega_A \subseteq \Omega$ be the elements $x \in \Omega$ on which $m_A(x) > m_B(x)$, and similarly let $\Omega_B \subseteq \Omega$ be the elements $x \in \Omega$ on which $m_B(x) > m_A(x)$. Finally let $\Omega_0$ be those elements $x \in \Omega$ for which $m_A(x) = m_B(x)$. Note that $\Omega_A$, $\Omega_B$, and $\Omega_0$ are disjoint, and that their union is the entire universe $\Omega$. We can then write the cardinality of the intersection of $A$ and $B$ as

$$\int_\Omega m_{A \cap B}(x) \, d\lambda(x) = \int_\Omega \min\{m_A(x), m_B(x)\} \, d\lambda(x)$$
$$= \int_{\Omega_A} m_B(x) \, d\lambda(x) + \int_{\Omega_B} m_A(x) \, d\lambda(x) + \int_{\Omega_0} m_A(x) \, d\lambda(x).$$

We then observe that $|A| + |B| - 2|A \cap B|$ is

$$\int_{\Omega_A} m_A(x) \, d\lambda(x) + \int_{\Omega_B} m_A(x) \, d\lambda(x) + \int_{\Omega_0} m_A(x) \, d\lambda(x)$$
$$+ \int_{\Omega_A} m_B(x) \, d\lambda(x) + \int_{\Omega_B} m_B(x) \, d\lambda(x) + \int_{\Omega_0} m_B(x) \, d\lambda(x)$$
$$- 2 \int_{\Omega_A} m_B(x) \, d\lambda(x) - 2 \int_{\Omega_B} m_A(x) \, d\lambda(x) - 2 \int_{\Omega_0} m_A(x) \, d\lambda(x).$$

Simplifying, we obtain

$$\int_{\Omega_A} m_A(x) - m_B(x) \, d\lambda(x) + \int_{\Omega_B} m_B(x) - m_A(x) \, d\lambda(x) = \int_{\Omega_A} |m_A(x) - m_B(x)| \, d\lambda(x).$$

But this is exactly the cardinality of the symmetric difference $|A \triangle B|$. ∎

## E  PROOF OF THEOREM 4.2.3

We will show that for any probabilistic universe transformation $\ell : \Omega \to \Delta(\mathcal{U})$, the induced expectation transformation $L : \Omega^* \to \mathcal{U}^*$ preserves cardinalities. That is, $|A| = |L(A)|$. (For easy of notation, we write $\mathcal{U}$ here rather than the $\hat{\mathcal{U}}$ used in statement of the result in the main text.)

*Proof.*  Recall that since $L(A)$ is a multiset and thus a measure, we have $|L(A)| = (L(A))(\mathcal{U})$. Expanding $L(A)$, we obtain

$$\left( \int_\Omega (\ell(x)) \, d\mu_a(x) \right) (\mathcal{U}) = \int_\Omega (\ell(x))(\mathcal{U}) \, d\mu_a(x) = \int_\Omega d\mu_a(x) = \mu_A(\Omega) = |A|,$$

where the second equality holds because $\ell(x)$ is a probability measure over $\mathcal{U}$. ∎

## F  NETWORK ARCHITECTURES

Given input images with $c$ channels, and an output dimension $d$, the function $\phi$ is parametrized by the network:

Table 1: Mean absolute errors in symmetric difference size prediction on MNIST

|  | sizes $\in [2, 5]$ | sizes $\in [2, 10]$ | sizes $\in [2, 20]$ |
|---|---|---|---|
| Multisets | 0.0722 | 0.1264 | 0.2061 |
| Multisets, learned op. | 1.1748 | 1.7159 | 4.0468 |
| Unrest. multisets | 0.5614 | 0.7693 | 1.1813 |
| Unrest. multisets, learned op, | 1.1402 | 1.7317 | 4.4426 |
| DeepSets | 0.6630 | 1.4097 | 4.0330 |
| DeepSets, learned op. | 1.2152 | 1.9386 | 5.2831 |

Table 2: Mean absolute errors in intersection size prediction on MNIST

|  | sizes $\in [2, 5]$ | sizes $\in [2, 10]$ | sizes $\in [2, 20]$ |
|---|---|---|---|
| Multisets | 0.0375 | 0.0648 | 0.1092 |
| Multisets, learned op. | 0.6021 | 0.8583 | 2.3487 |
| Unrest. multisets | 0.30644 | 0.4211 | 0.5708 |
| Unrest. multisets, learned op, | 0.6122 | 0.8750 | 2.5015 |
| DeepSets | 0.3583 | 0.6763 | 2.4277 |
| DeepSets, learned op. | 0.5258 | 0.8420 | 2.6967 |

1. A two-dimensional convolution layer with $c$ input channels, 20 output channels, kernel size 5, and stride 1 (no padding)

2. A ReLU activation

3. A two-dimensional max-pooling layer with kernel size 2 and stride 2

4. A two-dimensional convolution layer with 20 input channels, 50 output channels, kernel size 5, and stride 1 (no padding)

5. A ReLU activation

6. A two-dimensional max-pooling layer with kernel size 2 and stride 2

7. A fully-connected linear layer with output size $d$ (the input size is determined by $c$)

For the DeepSets model, we used for $\rho_1$ the architecture:

1. A fully-connected linear layer with input size $d$ and output size 100

2. A hyperbolic tangent activation

3. A fully-connected linear layer with input and output size 100

For $\rho_2$ we used:

1. A fully-connected linear layer with input and output size 100

2. A hyperbolic tangent activation

3. A fully-connected linear layer with input size 100 and output size 1

## G  ADDITIONAL FIGURES

Table 3: Accuracy on containment prediction for models trained on symmetric difference on MNIST

|  | sizes $\in [2, 5]$ |
|---|---|
| Multisets | 0.9653 |
| Multisets, learned op. | 0.2576 |
| Unrest. multisets | 0.6332 |
| Unrest. multisets, learned op, | 0.2679 |
| DeepSets | 0.6299 |
| DeepSets, learned op. | 0.2487 |

Table 4: Accuracy on containment prediction for models trained on intersection on MNIST

|  | sizes $\in [2, 5]$ |
|---|---|
| Multisets | 0.9633 |
| Multisets, learned op. | 0.2494 |
| Unrest. multisets | 0.4916 |
| Unrest. multisets, learned op, | 0.2819 |
| DeepSets | 0.5018 |
| DeepSets, learned op. | 0.3389 |

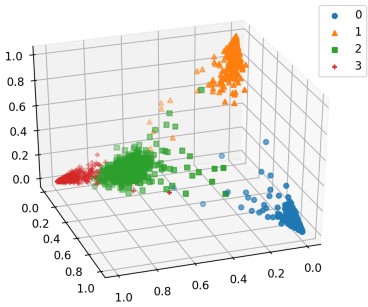 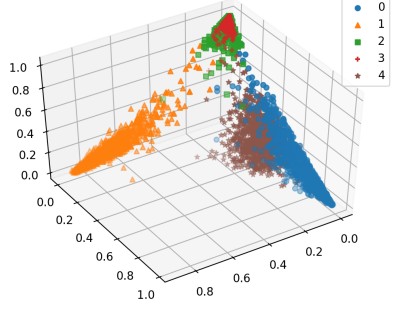

(a) Model trained on zeros, ones, twos, and threes.

(b) Model trained on zeros, ones, twos, threes, and fours.

Figure 2: Three-dimensional representations of test-set MNIST images generated by the restricted multiset model trained on multisets of sizes $\in [2, 5]$. Note that in (b) the representations of twos and threes are essentially inseparable.

## H  CLUSTERING $n$ OBJECTS GIVEN $n-1$ SYMMETRIC SET DIFFERENCE SIZES

We are interested in the following problem. Suppose we have a set of $n$ objects $\mathcal{U}$, each of which belongs to one of $k$ clusters, $C_1, \ldots, C_k$. Let $M : 2^{\mathcal{U}} \to \{1, \ldots, k\}^*$ be the function which takes any subset of $\mathcal{U}$, and gives the multiset of cluster labels represented in that subset. We are given oracle access to the function $\Delta : 2^{\mathcal{U}} \times 2^{\mathcal{U}} \to \mathbb{N}$ which gives the size of the symmetric set difference between the cluster-label multisets: $\Delta(A, B) = |M(A) \triangle M(B)|$. How many queries are required to determine the clusters $C_1, \ldots, C_k$ (up to permutation)?

We show that the clusters can be determined with $n-1$ specific queries. (Another way to think of this is as a training data problem, rather than an oracle querying problem; we show $n-1$ training examples can be sufficient.) We do this in two steps. The step lets us identify $k$ disjoint subsets of $\mathcal{U}$, such that no two of these subsets contain objects from the same cluster. The second step confirms that these subsets are in fact the clusters $C_1, \ldots, C_k$.

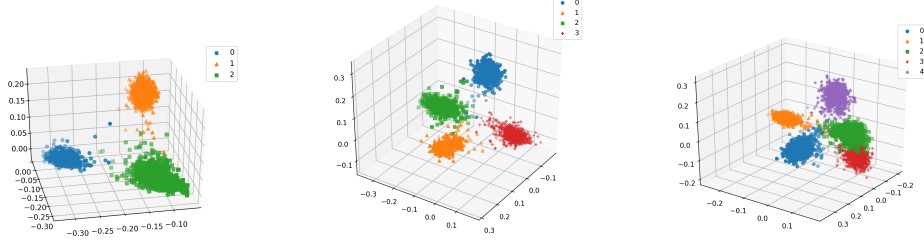

(a) Model trained on zeros, ones, and twos. (b) Model trained on zeros, ones, twos, and threes. (c) Model trained on zeros, ones, twos, threes, and fours.

Figure 3: Three-dimensional representations of test-set MNIST images generated by the unrestricted multiset model trained on multisets of sizes $\in [2, 5]$. Note that in (c), the clusters essentially form a tetrahedron, with one of the vertices being the combination of twos and threes.

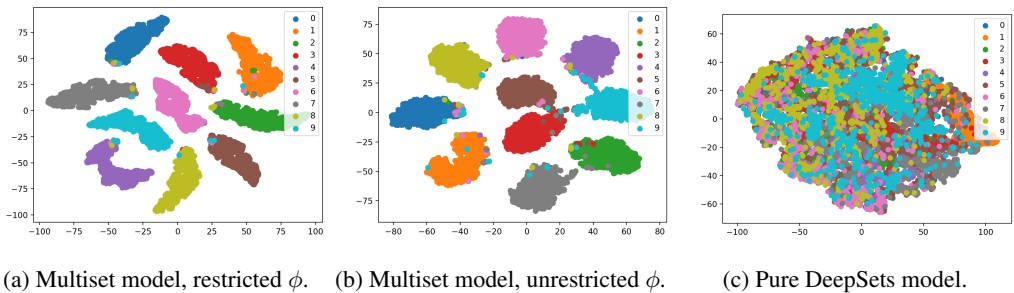

(a) Multiset model, restricted $\phi$. (b) Multiset model, unrestricted $\phi$. (c) Pure DeepSets model.

Figure 4: Two dimensional TSNE (van der Maaten and Hinton, 2008) "projections" of the ten-dimensional representations of test-set MNIST images generated by the models; the models were trained on multisets of sizes $\in [2, 5]$.

The first step consists of logarithmically "splitting" $\mathcal{U}$. The very first query in this step is $\Delta\left(\bigcup_{i=1}^{\lceil k/2 \rceil} C_i, \bigcup_{i=\lceil k/2 \rceil}^{k} C_i\right)$, which tells us that $\bigcup_{i=1}^{\lceil k/2 \rceil} C_i$ and $\bigcup_{i=\lceil k/2 \rceil}^{k} C_i$ are disjoint in terms of represented clusters. We proceed recursively, each query "splitting" the sets in half (in terms of which clusters they contain). The number of such steps required is $k - 1$ (which is the number of internal nodes in a balanced binary search tree for $k$ objects). We'll call the resulting disjoint sets $\tilde{C}_1, \ldots, \tilde{C}_k$ (since we technically don't yet know they correspond to the true clusters).

For the second step, we must verify that the objects in each of our sets resulting from step one all belong to the same cluster. This can be done by ordering the objects within each set, and comparing each consecutive pair as singletons. For each of our sets $\tilde{C}_i$, we thus make $|\tilde{C}_i| - 1$ such queries. Across all such sets, we thus make $\sum_{i=1}^{k} |\tilde{C}_i| - 1 = n - k$ queries.

So, the total number of queries made is $(k - 1) + (n - k) = n - 1$.

