# OpenReview forum: "Representation Learning with Multisets"
_ICLR.cc/2020/Conference — Reject_

### Official Review · AnonReviewer3 · 2019-10-15
**Official Blind Review #3**

**Rating:** 3

**Review:**

This paper proposes a new task of learning from sets, predicting the size of the symmetric difference between multisets, and gives a new method to solve the task based on the fuzzy set theory.
Although the topic of learning from sets is relevant and using the fuzzy set theory for the task is interesting, I have the following concerns regarding with the clarity, significance, and evaluation.

- Motivation is not clearly presented. The new task of predicting the size of the symmetric difference between multisets is proposed, while its application is not well discussed.
    Although Theorem 1 characterizes the task using the subset inclusion relationship, its relevance to applications is still not clear.
- The problem to be solved is not mathematically formulated. In particular, what are input and output?
- More detailed explanation of the data preparation would be required.
    How to transform images to pairs of multisets?
    Is the label (number) of each is used as an element of a multiset?
- For the second comparison partner, why is \Delta(A, B) defined as \rho_2(\Psi(A) + \Psi(B))?
    For fair comparison, this function should be the same with the proposed method, that is, ||\Psi(A) - \Psi(B)||_1 for the learned \Psi by DeepSets.
- In experiments, one of the most straightforward ways is to first predict labels for each image, followed by computing the symmetric difference from the predicted labels. Comparison with such baseline should be performed.

Minor comments:
- P.1, L.1 of the second paragraph: "The the" -> "The"


**Experience Assessment:**

I do not know much about this area.

**Review Assessment: Checking Correctness Of Derivations And Theory:**

I assessed the sensibility of the derivations and theory.

**Review Assessment: Checking Correctness Of Experiments:**

I carefully checked the experiments.

**Review Assessment: Thoroughness In Paper Reading:**

I read the paper at least twice and used my best judgement in assessing the paper.

---

> ### Author Response · Authors · 2019-11-15
> **Thank you for your comments (revision notes)**
>
> We thank you for your feedback on our paper!
>
> After reading your comments, we agreed that our motivation could be clearer. We have included a new perspective on the problem by defining it formally, which we hope addresses your concerns.
>
> As mentioned in our response above, we also agree that our experiments could better isolate our contributions – we have included new experiments to attempt to do so. We also tried to make our experimental methods a little clearer in "training and evaluation procedures."
>
> On the fact that this task is easy if you know the labels for individual objects: we agree 100%. The reason we think our problem is interesting is because we do not have access to such labels, but still want to about the structure they exhibit. Again, we think and hope our formalization of the problem in our revision helps make this clear.

---

### Official Review · AnonReviewer2 · 2019-10-22
**Official Blind Review #2**

**Rating:** 3

**Review:**

This paper presents a framework for learning representations of multisets.  The approach is built on top of the DeepSets (Zaheer et al., 2017) model, which already is capable of learning representations of multisets, with a new objective of predicting the size of symmetric differences between two multisets, plus a small change in the implementation of the DeepSets model by restricting the per-element module to output a point in the probability simplex.

I liked the background on fuzzy sets and the development of fuzzy multisets and different operations on these multisets.  The definitions and arguments are quite clear and helpful.  One small suggestion for page 4 is that I can understand why the formulation is the only sensible choice for multisets with desired properties, but a claim like this deserves a proper proof.

Model-wise the paper made two contributions for learning representations for multisets as mentioned above: (1) proposed the symmetric difference prediction as a task for learning representations, the argument for this task is that predicting symmetric difference implicitly encourages the model to learn about containment; (2) a slight change in the DeepSets model architecture where the outer rho function is identity and the inner phi function has to output a point in the simplex.

I found these technical contributions to be a bit small.  In addition to this, the paper only presents results on MNIST in a toyish setting, this makes me feel the paper may be more suited for publication in a workshop (idea is interesting, small scale experiments to illustrate the insights, but not complete enough to be published at a conference).

Regarding contribution (1), I can see why predicting symmetric difference makes sense as argued in the paper, but I’m not convinced that this is better than other alternatives.  In order to show that this is a reasonable approach for learning representations, some results that compare this with other possible learning objectives would be necessary.  But I don’t see any such results in this paper.

Regarding contribution (2), I feel the restriction of the phi function to output points in simplex is not very well motivated and confusing in the first read.  Again I can understand why we may want to do this but don’t see why we need to do this.  I’m also concerned that such an architecture may only be good for the task of predicting symmetric difference as it is customized for this task.  Figure 3 shows that an unrestricted model seems to learn better representations despite a worse symmetric difference prediction error, which again confirms the concern.

Another thing about the experiment setup: the second baseline, labeled “DeepSets” in Table 1 actually changed two things compared to the proposed approach: (1) changing the psi function and (2) also changed the symmetric difference function.  It would be good to isolate the contribution of the two.

Overall I feel this paper is not yet ready to be published at ICLR.

**Experience Assessment:**

I have published one or two papers in this area.

**Review Assessment: Checking Correctness Of Derivations And Theory:**

I carefully checked the derivations and theory.

**Review Assessment: Checking Correctness Of Experiments:**

I carefully checked the experiments.

**Review Assessment: Thoroughness In Paper Reading:**

I read the paper thoroughly.

---

> ### Author Response · Authors · 2019-11-15
> **Thank you for your comments (revision notes)**
>
> Thank you for your thoughtful comments!
>
> We have revised our work to address your concerns as much as possible.
>
> To your first point, we agree that our definitions of multiset operations could be better motivated. We have added a formalization of the problem we are trying to solve and of multisets themselves, which we hope address your concerns here. We hope that our additions also make it clear that we see our contributions not just as modifications to DeepSets functions, but as fundamentally interesting/new ways of looking at containment relations. (This formalization also explains the restriction of \phi to the probability simplex.)
>
> As for your suggestion of comparisons with other objectives, we have included prediction the size of the intersection as a possible task, and compare it experimentally to the symmetric difference. It is not totally clear from the theory what other alternatives there are (although we're sure they exist!) We see our contribution here as an interesting first step in exploring the problem which we've formalized. From the theoretical perspective we lay out in our revision, however, we think the symmetric difference (and intersection) is well motivated.
>
> Finally, we agree 100% that our baselines would be more interesting if we tried harder to isolate different aspects of our model. Our revised paper includes these experiments.

---

### Official Review · AnonReviewer1 · 2019-10-23
**Official Blind Review #1**

**Rating:** 6

**Review:**

Authors of this paper propose train a model by predicting the size of the symmetric difference between pairs of multisets. With the motivation from fuzzy set theory, both the multiset representation and predicting symmetric difference sizes given these representations are formulated.

In Section 3.3, authors stated that theorem 3.3.1 provides the compelling reason to use symmetric difference over intersection or non-symmetric difference. The statement seems not so straightforward, and how it works as the learning criterion for semi-supervised clustering in the experiments.

For the relaxation used for the normalization of \phi(a), does this restrict the feasible space of the standard basis vectors? In Section 4.3, authors claimed that in the case of n=3, 98.9% classification accuracy can be obtained by simply picking the maximum valued coordinate of the representation of each object. A systematic comparison in terms of the classification accuracy is important for evaluating the semi-supervised clustering problem.

In Section 4.2, authors directly model the mean absolute errors in symmetric difference size prediction. It might be more interesting to see what real problems the proposed model can naturally be applied.

**Experience Assessment:**

I have read many papers in this area.

**Review Assessment: Checking Correctness Of Derivations And Theory:**

I assessed the sensibility of the derivations and theory.

**Review Assessment: Checking Correctness Of Experiments:**

I assessed the sensibility of the experiments.

**Review Assessment: Thoroughness In Paper Reading:**

I read the paper at least twice and used my best judgement in assessing the paper.

---

> ### Author Response · Authors · 2019-11-15
> **Thank you for your comments**
>
> Hi! Thank you for your thoughtful feedback.
>
> We have taken it into as much consideration as possible in our revision.
>
> Our main revision is a formalization of the problem we are trying to solve. We hoped to make clear here that the symmetric difference is not really a learning criterion for semi-supervised clustering. Rather, the error on predicting symmetric difference relates directly to how well the model captures the desired notion of "containment" (which we state more formally in paper).
>
> We are not 100% sure whether this answers your question about the normalization of \phi, but we provide a formal justification for this, by relating \phi to a probability distribution, i.e. a point in the probability simplex.
>
> We agree that a more nuanced look at the classification accuracy (up to permutation)  obtained by the semi-supervised clustering would be helpful. As this wasn't the main focus of our paper, we hope that instead our new experiments argue for the usefulness of the learned representations.
>
> Finally, we do mention a possible real-world application we think our model could apply to during our exposition of the problem. While, this is mentioned more as a motivation in this paper for how we define the problem, we are definitely excited about possible future applications of our methods.

---

### Decision · Program_Chairs · 2019-12-19

**Decision:**

Reject

**Comment:**

While the reviewers appreciated the problem to learn a multiset representation, two reviewers found the technical contribution to be minor, as well as limited experiments. The rebuttal and revision addressed concerns about the motivation of the approach, but the experimental issues remain. The paper would likely substantially improve with additional experiments.